# The Risk of Malignant Degeneration of Müllerian Derivatives in PMDS: A Review of the Literature

**DOI:** 10.3390/jcm12093115

**Published:** 2023-04-25

**Authors:** Federica Gagliardi, Augusto Lauro, Livia De Anna, Domenico Tripodi, Anna Esposito, Flavio Forte, Daniele Pironi, Eleonora Lori, Patrizia Alba Gentile, Ignazio R. Marino, Ernesto T. Figueroa, Vito D’Andrea

**Affiliations:** 1Department of Surgery, Sapienza University of Rome, 00161 Rome, Italy; 2Urology Department, M.G. Vannini Hospital, 00177 Rome, Italy; 3Sidney Kimmel Medical College, Thomas Jefferson University, Philadelphia, PA 19107, USA; 4Division of Pediatric Urology, Nemours/Alfred I. DuPont Hospital for Children, Wilmington, DE 19803, USA

**Keywords:** cryptorchidism, persistent Müllerian duct syndrome (PMDS), malignant degeneration, anti-Müllerian hormone (AMH), orchipexy

## Abstract

Persistent Müllerian Duct Syndrome (PMDS) is a rare autosomal recessive disorder of sex development characterized by the presence of fallopian tubes, uterus and upper one-third of the vagina in individuals with XY genotype and normal male phenotype. The main complications of PMDS are infertility and the rare risk of malignant degeneration of both testicular and Müllerian derivatives. We report the case of a 49-year-old man who, during repair of an incisional hernia, was incidentally found to have a uterine-like structure posterior to the bladder. In the past at the age of 18 months, he had undergone bilateral orchidopexies for bilateral cryptorchidism. The intraoperative decision was to preserve the uterine-like structure and make a more accurate diagnosis postoperatively. Evaluation revealed an XY chromosome and imaging consistent with PMDS. The patient was informed about the risk of neoplastic transformation of the residual Müller ducts and was offered surgical treatment, which he declined. Subsequent follow-up imaging studies, including testicular and pelvic ultrasound, were negative for findings suggestive of malignant testicular and Mullerian derivative degeneration. A review of the international literature showed that, when a decision is taken to remove the Mullerian derivatives, laparoscopy and especially robotic surgery allow for the successful removal of Müllerian derivatives. Whenever the removal of these structures is not possible or the patient refuses to undergo surgery, it is necessary to inform the patient of the need for adequate follow-up. Patients should undergo regular pelvic imaging examination and MRI might be a better method for that purpose.

## 1. Introduction

Persistent Müllerian Duct Syndrome (PMDS) is a rare autosomal recessive disorder of sex development characterized by the presence of residual Müllerian structures in the male. Some of the structures may include fallopian tubes, uterus and a portion of a vagina in individuals with XY genotype and normal male phenotype. These structures are often incompletely formed and rudimentary. The syndrome is caused either by a mutation in the anti-Müllerian hormone (AMH) gene or the AMH receptor gene, leading to a failure of formation or action of AMH in intrauterine life [1].

### 1.1. Sex Determination and Differentiation: An Overview

The sex of the embryo is established at the moment of fecundation, but gonads acquire male or female characteristics at 7-week gestation starting from an undifferentiated bipotential gonad. Starting from this bipotential gonadal primordium, sexual development takes place through two distinct but sequential processes: “sexual determination” which directs the sex-specific differentiation of the gonad according to the chromosomal sex established at the time of fecundation; the “sexual differentiation”, which depends on the presence or absence of the testicle, characterized by the development of internal and external genitalia [2].

In males, the process of sexual development initially involves the formation of the testis starting from the bipotential gonadal primordium (sexual determination), subsequently, the formation of a male’s internal and external genitalia (sexual differentiation) is accomplished thanks to the hormone-secretive action of the fetal testis [2].

The SRY (Sex-determining Region Y) gene, located on the short arm of the Y chromosome, is the testis-determining gene that initiates male sex determination. Its activation, in 46, XY embryos, is associated with a peculiar pattern of gene expression and induces the progressive morphological and functional differentiation of the bipotential gonad into testis and, at the same time, inhibits the development of the female gonad [3].

The further development of the male reproductive system (sexual differentiation) is influenced by hormones produced by the fetal testis: the anti-Müllerian hormone (AMH) produced by Sertoli cells and the testosterone produced by Leydig cells [4].

As in the gonads, the process of sex-specific differentiation of the genital ducts is preceded by the development, in both XX and XY embryos, of sexually undifferentiated primordia. These structures, which originate from the mesonephros at 8-week gestation, are represented by two equal and parallel ducts: the mesonephric duct of Wolff and the para-mesonephric duct of Müller [5].

The AMH synthesis by Sertoli cells causes the progressive degeneration and disappearance of the Müller ducts. This phenomenon, which takes place between 7- and 10-week gestation, represents the first sign of male differentiation of the internal genitalia, determining the regression of those structures which in the female embryo will instead give rise to the fallopian tubes, uterus, cervix and proximal third of the vagina [6].

Conversely, testosterone, which is produced by Leydig cells starting from the eighth week, positively influences the development of Wolff’s ducts, which differentiate into efferent ducts of the testis, epididymis, vas deferens, seminal vesicles and ejaculatory ducts.

The external genitalia, which are also initially identical in all fetuses regardless of chromosomal or gonadal sex, develop, from the tenth week of gestation until birth, in the presence of adequate quantities of dihydro-testosterone (DHT), formed by the reduction of testosterone by the enzyme 5-alpha-reductase [5]. So, the prostate is formed and the progressive masculinization of the external genitalia takes place. During the last trimester, testis migrate through the inguinal canal in the scrotum, partly guided by testosterone and the Insulin-Like Factor 3 (INSL3) produced by the Leydig cells. DHT is also responsible for the development of secondary sexual characteristics in normal males from puberty onwards.

### 1.2. Anti-Müllerian Hormone

The anti-Müllerian hormone (AMH) is a 140 kDa homodimeric glycoprotein belonging to the TGF-β superfamily [7].

Basal hormone secretion is gonadotropin-independent, but is regulated by specific transcription factors: the first factor able to be bound to the promoter of the AMH gene and to induce its expression is the transcription factor SOX9; subsequently, other factors such as SF1, GATA4 and WT1 guarantee the increase of its biosynthesis [8].

Once the secretion of AMH has begun, its further production is supported by the action of FSH which induces the proliferation of Sertoli cells and determines an up-regulation of the transcription of the AMH encoding gene [8].

The serum concentration of the anti-Müllerian hormone remains stable and high (with reference values ranging between a minimum of 55 ng/mL and a maximum of 250 ng/mL in children aged two to nine) until the onset of puberty, when it begins to gradually decline [8].

This down-regulation in AMH secretion is androgen dependent: it is associated with an increase in the intratesticular concentration of testosterone responsible for morphological and functional modifications of Sertoli cells which from immature and proliferating cells become mature and quiescent cells [8].

Therefore, at the beginning of pubertal development, the increase in testosterone determines a maturation/differentiation of Sertoli cells and a consensual down-regulation of the expression of the AMH gene which results in a rapid and progressive decrease in the values of the Sertolian hormone.

If in the adult the evaluation of the endocrine function of the hypothalamic-pituitary-gonadal axis is essentially based on the determination of the serum levels of gonadotropins, testosterone and inhibin B, in the pediatric age the serum concentration of the anti-Müllerian hormone constitutes a precious biological marker that provides information about the activity of the gonad and about the action that FSH and androgens play on it.

During childhood, the anti-Müllerian hormone together with inhibin B constitutes the only marker of Sertolian function; on the contrary, after puberty, and in adult men, the significant decrease in AMH values does not allow its diagnostic use.

## 2. PMDS

### 2.1. Genetics and Molecular Aspects

It is estimated that about 45 percent of cases of PMDS are caused by mutations in the AMH gene, and referred as PMDS type 1 (PMDS1). Among the remaining patients, about 40 percent of cases are caused by mutations in the AMH type 2 receptor (AMHR2) gene, referred as PMDS type 2 (PMDS2).

(1)PMDS1: the anti-Müllerian hormone gene is located on the short arm of chromosome 19 (19p13.3), has a length of 2.8 kb and is formed by five exons coding for a 535 amino acid protein, it is a member of the transforming growth factor beta (TGF-beta) family [9]. Several different AMH gene mutations have been identified: deletions, insertions, missense, nonsense and splicing mutations play a pathogenetic role in the determinism of the disease. The effect of mutations on the molecule function is heterogeneous: some prevent an adequate post-translational folding of the C-terminal region and this determines a rapid degradation of the molecule, even before its secretion; others result in the biosynthesis of a truncated protein in which the C-terminal portion, the one with biological activity, is missing; moreover others are associated with the synthesis and secretion of a hormone that has lost the ability to bind to its receptor [10]. As a consequence of this spectrum of mutations affecting the AMH gene, the AMH serum levels are characteristically low or undetectable in prepuberal patients with PMDS type I. The diagnostic value of AMH serum concentration is lost in adults in whom the hormone is physiologically undetectable.(2)PMDS2: the action of AMH on target tissues is made possible by the expression on them of a heterodimeric receptor consisting of AMH type 1 (AMHR1) and type 2 receptor (AMHR2). Only the type 2 receptor is AMH-specific and it is mutated in PMDS type 2 patients. It is a transmembrane receptor of 573 amino acid consisting of an N-terminal extracellular domain that binds AMH, a single transmembrane domain, and a C-terminal intracellular domain exhibiting serine/threonine kinase activity [10]. The AMHR2 is an 8 kb gene located on the long arm of chromosome 12 (12q13), consisting of 11 exons. The first three exons code for the extracellular domain that binds the hormone, exon four codes for the transmembrane domain, while the remaining seven exons code for the intracellular domain associated with the catalytic activity of the receptor. All exons can be affected by mutations; as for the AMH gene, it can be of various natures: missense, nonsense, deletions and splicing mutations have been detected [10]. Specifically, the mutation most frequently found in patients with PMDS2 is a 27-base pair deletion affecting exon 10 [1]. Mutations in the AMHR2 gene prevent the AMH from carrying out its action on target tissues. The insensitivity of the Müllerian ducts to AMH therefore means that in the fetus, they do not undergo the physiological process of involution but give rise to the development of the fallopian tubes, uterus and proximal third of the vagina. In contrast to type 1 PMDS, where the syndrome develops as a consequence of the receptor mutation, the serum concentration of AMH is normal in prepuberal males.

About 12% of PMDS patients do not have mutations in either AMH gene or the gene coding for its type 2 receptor [10]. Although it cannot be excluded that this form is associated with unidentified mutations of the aforementioned genes, some clinical characteristics of idiopathic PMDS (for example, never presenting with transverse testicular ectopia) allow us to make the hypothesis that a different genetic pathway is involved in these cases. Multiple genetic pathways involved in the development of Müller’s ducts have been implicated, but most notably the Wnt/beta-catenin system [10]. The study of the genetic alterations inherent in the Wnt/beta-catenin pathway in the future could therefore shed light on the genetic basis of those forms of PMDS which today are considered idiopathic.

### 2.2. Clinical Features

The main clinical presentation of PMDS is cryptorchidism, in association with the often unexpected finding of residual Müllerian structures. Taking into account the anatomical location of testes and Müllerian derivatives, and the unilateral or bilateral nature of cryptorchidism, three main clinical presentations of PMDS have been described [10]:(1)Bilateral Cryptorchidism: the testes are both located in the pelvis assuming a position similar to the one normally presented by the ovaries; they are included in the broad ligament of the uterus, close to the fimbriae of the fallopian tubes. This is estimated to occur in 60–70% of patients [11].(2)Unilateral Cryptorchidism: one of the two testicles are located in the pelvis while the contralateral one is in the scrotum. The descended testicle is associated with the presence of an ipsilateral inguinal or inguinal-scrotal hernia whose content is represented by the fallopian tube of the same side and the uterus, so that this clinical presentation is often referred as a “hernia uteri inguinalis”. This is estimated to occur in 20–30% of patients [12].(3)Transverse Testicular Ectopia: this is the most specific but least common clinical presentation of PMDS and is characterized by the presence, in a single hemiscrotum, of a hernia sac containing both testes, both tubes and the uterus [12].

The most significant complications of PMDS are infertility and the rare risk of malignant degeneration of both testicular and Müllerian derivatives.

Cryptorchidism, injury of the excretory ducts or impaired testicular blood flow- which can both occur during orchipexy and/or the surgical removal of the Müllerian derivatives are the most important causes of infertility. Cryptorchidism is considered one of the main risk factors for testicular malignant degeneration. The incidence of testicular malignant change in a patient with PMDS is the same of a cryptorchid man without PMDS [13]. Malignant degeneration of Müllerian derivatives is much less frequent but at least 12 cases are reported in the literature.

The management of the residual Müllerian structures remains controversial, as their surgical excision has been associated with inadvertent injury to the adjacent Wolfian structures. We herein describe a case report with review of the literature reporting malignant cases in patients with PMDS.

## 3. Case Report

A 49-year-old male, married but with no children, came to our hospital reporting a 20-year presence of swelling in the supraumbilical area of the abdominal wall. The swelling was neither painful nor tender, and corresponded to an incisional hernia resulting from a previous laparotomy for intestinal perforation. In his past medical history, we recorded orchidopexy for bilateral cryptorchidism at the age of 18 months, surgery for a left varicocele at the age of 26 years and emergency laparotomy at the age of 27 years for intestinal perforation of an unspecified cause and nature. The abdominal examination revealed the presence of a xipho-pubic surgical scar where there was a reducible ventral hernia in the supraumbilical area, with a maximum diameter of 15 cm, tense-elastic consistency. Testicles were found in the scrotum, with a volume of 6 mL on the left and 7 mL on the right and a reduced consistency. On the right side, there was a surgical scar as a result of previous surgery for cryptorchidism. The dimensions of the penis were within limits (12 cm × 4 cm). Preoperative imaging consisted of a CT scan limited to the upper abdominal wall, showing 9 × 11 cm diastasis of the anterior rectus abdominal muscles. The decision was made to proceed with surgical repair of the ventral hernia. Our choice to perform a laparotomy instead of laparoscopy was related to the patient’s personal history: he had an emergency laparotomy for intestinal perforation and post-operative peritoneal adhesions could have made surgery more challenging and less safe if performed laparoscopically. During the surgery, a uterine-like structure of parenchymatous consistency was found posterior to the bladder (Figure 1).

Due to the unexpected finding and the lack of informed consent, the decision was made to preserve the uterine-like organ and proceed with additional evaluation to make a more accurate diagnosis. The surgical findings were discussed with the patient, and he consented to a further diagnostic examination with a contrast enhanced MRI of the lower abdomen. The MRI showed the presence of an 8.5 × 2 cm cylindrical, solid lesion, located above the bladder, similar to a normoflexed, rudimentary, dysmorphic uterus, probably originating from the prostatic urethra (Figure 2).

Testicular echo-color-doppler was performed, confirming the mild bilateral testicular atrophy detected on physical examination, but no lesions suggestive of malignancy. The karyotype analysis showed a male karyotype 46XY. Based on the intraoperative findings, history of bilateral cryptorchidism, MRI imaging and karyotype, diagnosis of PMDS was made. The patient was counseled about the rare risk of neoplastic transformation of the residual Müller ducts and was offered surgical treatment to remove these structures, which he declined. The subsequent imaging follow-up with testicular and pelvic ultrasound were negative for malignant testicular and Müllerian derivative degeneration.

## 4. Literature Review and Discussion

PMDS becomes clinically apparent typically during exploration of children with non-palpable intra-abdominal undescended testes, or inguinal testes associated with an inguinal hernia. Since it was first described in 1939 by Nilson, approximately 300 cases of PMDS have been reported in the literature [14]. It is very likely that this figure underestimates the true incidence of the syndrome as the diagnosis of PMDS largely depends on the intraoperative detection of Müllerian derivatives. With the exception of cases of transverse testicular ectopia and palpable testes, the disease is rarely diagnosed preoperatively in patients who present with bilateral non-palpable (intra-abdominal) testes or those with unilateral undescended testes and inguinal hernias. However, PMDS should be suspected in the clinical setting of bilateral cryptorchidism or unilateral cryptorchidism associated with inguinal hernia.

The suspicion of the disease would allow targeted instrumental examinations (pelvic ultrasound and abdominal magnetic resonance imaging) to detect and identify Müllerian derivatives and, through karyotype analysis, to make a definite and early diagnosis of fundamental importance considering the rare risk of malignant degeneration of testicles and Müllerian remnants [13]. Although this condition is often diagnosed and treated in childhood, adult cases are not rarely reported [15]. The patient must be reassured about his sexual identity and it must be explained that the pathology does not affect normal sexual function, although it is often associated with infertility. The patient must be made aware that Müllerian derivatives have an estimated 8% risk of neoplastic degeneration [15], and it is recommended to remove them whenever possible. On the other hand, preservation of Müllerian structures [16] and follow–up is a possibility when the patients are not keen to undergo a further surgical procedure: in the literature [17,18,19,20,21,22,23,24,25,26,27,28], only 12 patients out of almost 300 cases presented malignant degeneration between the age of 4 and 68 years (Table 1).

In addition to the risk of Müllerian derivatives degeneration, PMDS patients show an increased risk of developing testicular neoplasms with an incidence rate comparable to that of cryptorchid patients without PMDS [13]. Cryptorchidism and risk of malignant degeneration of the testes are clinical features detectable in another form of male pseudohermaphroditism called androgen insensitivity syndrome (AIS). AIS is a rare X-linked recessive genetic condition characterized by mutations of the androgen receptor gene: the partial or total lack of sensitivity of androgen receptors to androgen stimulation alters the masculinization process in chromosomal male patients (46XY karyotype) [29]. As a result of different mutations, the phenotype can range from female phenotype with well-developed external genitalia, a short, blind-ending vagina and cryptorchidism (the undescended testes can be located in abdomen, inguinal canals or labia majora) to male phenotype, with perineal hypospadias, micropenis and cryptorchidism [30]. The management of patients with Disorders of Sex Development as PMDS or AIS requires an adequate urologic surveillance.

The restoration of testes, the preservation of fertility, and the prevention of malignancy are the purposes of the surgical treatment in PMDS patients. Before evidence of the risk of malignant degeneration of Müllerian derivatives emerged, the surgical approach to the PMDS patients was aimed solely at protecting fertility and reducing the risk of developing testicular tumor. Orchidopexy and inguinal hernioplasty/alloplasty were not combined with simultaneous removal of the tubes, uterus and proximal portion of the vagina. This choice of the surgeons was justified by the technical difficulties encountered in the dissection of the vas deferens from the uterine wall: the close anatomical proximity between these two structures makes the risk of severe and irreversible damage of the vas deferens high. Although the risk of damaging the deferens duct or the deferential artery cannot be eliminated, laparoscopy and especially robotic surgery [16,31] allows a successful removal of Müllerian derivatives due to the high level of surgical precision guaranteed by these two techniques. It is imperative to inform the patient that surgery to remove Müller’s derivatives may itself lead to complications in the structures of the urinary genital tract, causing infertility in particular, but that removal of residual Müller’s duct is desirable given the risk of neoplastic degeneration, currently estimated at 8% [15].

## 5. Conclusions

When a decision is taken to remove the Müllerian derivatives, laparoscopy and especially robotic surgery allow a successful removal of Müllerian derivatives [16,31,32]. Whenever removal of these structures is not possible or the patient refuses to undergo surgery, it is necessary to inform the patient of the need for adequate follow-up [33]. Specifically, it is known that cryptorchidism is one of the main risk factors for testicular cancer, so monthly self-palpation of the testicles is recommended to potentially facilitate early cancer detection [34]. Patients with intact Müllerian structures should undergo pelvic imaging examination annually and MRI might be a better method than US for that purpose [33]. In addition, since the disease is genetically determined, siblings of affected patients should also undergo appropriate instrumental diagnostic investigations.

## Figures and Tables

**Figure 1 jcm-12-03115-f001:**
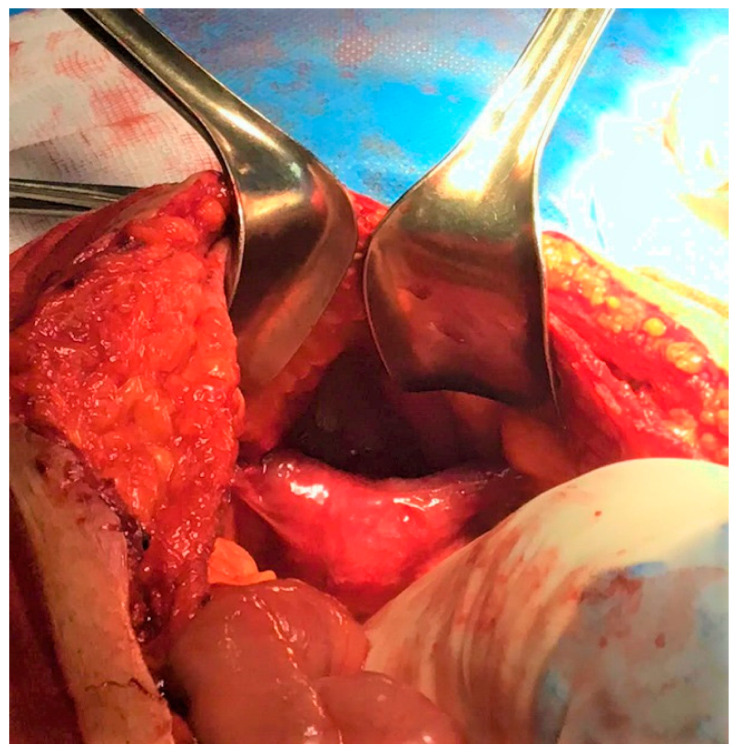
Intraoperative finding: unexpected uterine-like structure.

**Figure 2 jcm-12-03115-f002:**
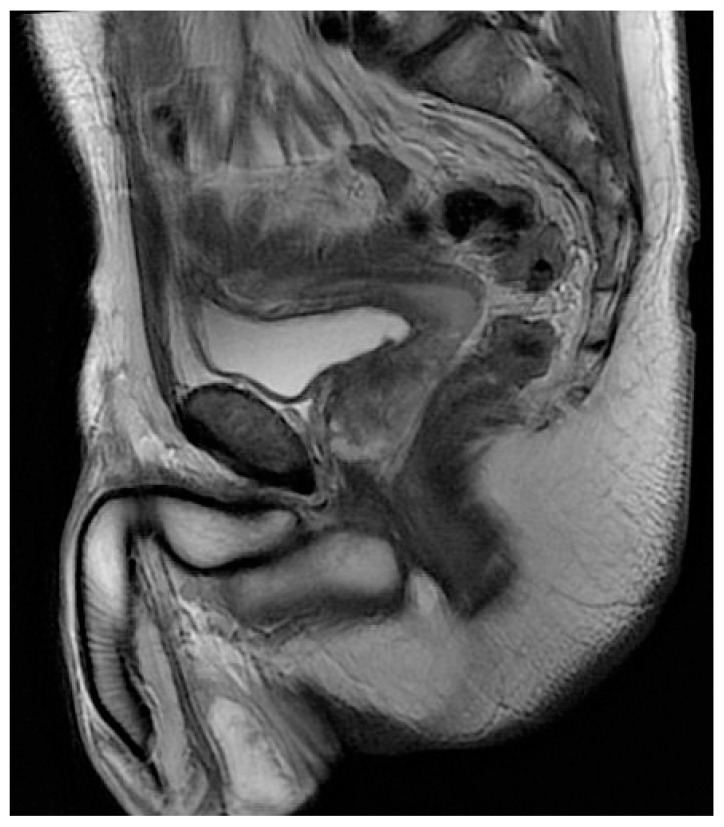
MRI confirmed the presence of a solid formation, located above the bladder, similar to a rudimentary uterus.

**Table 1 jcm-12-03115-t001:** Cases of malignant degeneration of Mullerian derivatives: literature review.

Reference	Case *n*.	Year	Age at Diagnosis (Years)	Previous Orchidopexy or Other Relevant Procedures Before Diagnosis	Presentation	Type of Müllerian Malignancy	Outcome
[17]	1	1968	44	At age 18 hypospadias repair, bilateral inguinal exploration for cryptorchidism and left orchidopexy.	Recurrent UTIs, Back pain, Urethral discharge	Squamous cell carcinoma	Inoperable tumor Metastatic spread Death
[18]	2	1976	68	none	Lower abdominal pain, Irritative bladder symptoms	Papillary cystadenocarcinoma	Resection of tumor Radiotherapy and estrogen therapy No evidence of recurrence at age 82
[19]	3	1981	33	Several years of hematuria: Cystoscopy, retrograde studies.	Hematuria, Right flank pain	Clear cell carcinoma	Resection of tumor No metastases, no further information given regarding recurrence
[20]	4	1990	4	none	UTI, Hematuria	Squamous cell carcinoma	Resection of tumor No metastases
[21]	5	1992	50	none	Hematuria	Adenocarcinoma	Resection of tumor Recurrence of tumor 4 years post-op Radiotherapy given No further information
[22]	6	1996	36	Radical surgery for hypospadias.	Lower abdominal pain, Fever	Squamous cell carcinoma	Metastatic spread Resection of tumor and bilateral cutaneous ureterostomy No further information given
[23]	7	2002	67	none	Autopsy	Clear cell adenocarcinoma	Pt died in traffic accident Adenocarcinoma found incidentally Metastasized to retroperitoneal lymph nodes and lungs
[24]	8	2005	14	Left nephrectomy at 2 months for multicystic kidney. Right orchiectomy at age 8, simultaneous left side inguinal orchidopexy for inguinal testicle.	Hematuria, Increasing lower abdominal protuberance	Adenosarcoma	Tumor resected 2 months post-op. lung, bone and abdominal metastases Death
[25]	9	2005	39	none	Abdominal pain, Hematuria, Urinary retention, Bilateral cryptorchidism	Endocervical adenocarcinoma	Tumor partially resected Developed liver metastases Died 1.5 years after first operation
[26]	10	2006	44	none	Hemospermia, Hematuria, Infertility for 15 years	Papillary cystadenocarcinoma	Not known
[27]	11	2006	15	none	Lower abdominal protrusion, Urinary retention	Clear cell adenocarcinoma	Resection of tumor No recurrence 16 weeks post-op
[28]	12	2017	45	Right inguinal hernia repair at 8 years	Painless hematuria, lower abdominal protuberance	Clear cell adenocarcinoma	Resection of the tumor Adjuvant radiotherapy to the pelvis (positive surgical margins) 3 months post-op lung metastases

UTIs: urinary tract infections.

## Data Availability

Not applicable.

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
