# Peer review of "The Risk of Malignant Degeneration of Müllerian Derivatives in PMDS: A Review of the Literature"

_jcm, 2023, doi:10.3390/jcm12093115_

Round 1

Reviewer 1 Report

Dear Authors,

Thank you for presenting a very interesting case report-

I have a question regarding the type of operative procedure, why did you choose laparotomy?

How often do you control the patient regarding the risk of neoplastic transformation?

Thank you,

Reviewer

Author Response

Point 1: I have a question regarding the type of operative procedure, why did you choose laparotomy?

Response 1:  Our choice to perform laparotomy instead of laparoscopy was related to the patient’s personal history: he had emergency laparotomy at the age of 27 years for intestinal perforation and post-operative peritoneal adhesions could have made surgery more challenging and less safe if performed laparoscopically.

Point 2: How often do you control the patient regarding the risk of neoplastic transformation?

Response 1: Patients in whom the Müllerian derivatives are not removed must undergo pelvic examination annually.

Reviewer 2 Report

The authors should be congratulated for how they present the rationale and clinical implications of this very interesting case with persisitent Mullerian duct syndrome (PMDS). despite it is only a case report I find the case is very interesting and presents the variable presentation of this entity and also the serious circumstance that it implies risk of malignancy, and how that should be prevented.

I would simply propose in the discussion to add one or two paragraphs regarding the analogies and disimilarities with another very interesting condition, that although rare is a bit more comon, the female pseudohermafroditism due to androgen insensitivity. A mention of this other very interesting entity will help differentiate it from PMDS and, on my opinion will enrich a little more the discussion. In fact, this other entity also shares the clinical circumstance of both gonadal malignancy and is also an inheritable entity.

Author Response

Point 1: I would simply propose in the discussion to add one or two paragraphs regarding the analogies and disimilarities with another very interesting condition, that although rare is a bit more comon, the female pseudohermafroditism due to androgen insensitivity. A mention of this other very interesting entity will help differentiate it from PMDS and, on my opinion will enrich a little more the discussion. In fact, this other entity also shares the clinical circumstance of both gonadal malignancy and is also an inheritable entity.

Response 1: Dear reviewer

Thank you for your suggestions. We enriched the discussion spending few words on androgen insensitivity syndrome.